# Hepatitis C Virus Epidemiology in Lithuania: Situation before Introduction of the National Screening Programme

**DOI:** 10.3390/v14061192

**Published:** 2022-05-30

**Authors:** Egle Ciupkeviciene, Janina Petkeviciene, Jolanta Sumskiene, Gediminas Dragunas, Saulius Dabravalskis, Edita Kreivenaite, Tadas Telksnys, Gediminas Urbonas, Limas Kupcinskas

**Affiliations:** 1Health Research Institute, Faculty of Public Health, Lithuanian University of Health Sciences, LT47181 Kaunas, Lithuania; janina.petkeviciene@lsmuni.lt (J.P.); tadas.telksnys@lsmuni.lt (T.T.); 2Department of Gastroenterology, Lithuanian University of Health Sciences, LT50161 Kaunas, Lithuania; jolanta.sumskiene@lsmuni.lt (J.S.); edita.kreivenaite@lsmu.lt (E.K.); limas.kupcinskas@lsmuni.lt (L.K.); 3Jurininku Health Care Centre, LT91213 Klaipeda, Lithuania; g.dragunas@jspc.lt (G.D.); s.dabravalskis@jspc.lt (S.D.); 4Department of Family Medicine, Lithuanian University of Health Sciences, LT50161 Kaunas, Lithuania; gediminas.urbonas@lsmu.lt

**Keywords:** hepatitis, HCV, epidemiology, screening, risk factors

## Abstract

In 2022, the Lithuanian health authorities decided to pay general practitioners a fee for performing serological tests for hepatitis C virus (HCV) antibodies in the population born from 1945 to 1994 once per life and annual HCV testing for PWID and HIV infected patients. This study aimed to assess trends in HCV-related mortality in the country and the prevalence of HCV infection among patients with liver diseases and evaluate possibilities of screening for HCV infection at a primary health care center. Age-standardized mortality rates in 2010–2020 were calculated for deaths caused by chronic hepatitis C and some liver diseases. Data on HCV infection among patients with liver cirrhosis, cancer and transplant patients were collected from the tertiary care hospital Kauno Klinikos. The prevalence of anti-HCV and risk factors of HCV infection was assessed among patients registered with the health care center in Klaipeda, where a pilot study of screening was performed. No steady trend in mortality was observed. Analysis of medical documentation showed that 40.5% of patients with liver cirrhosis, 49.7% with cancers and 36.9% of transplant patients were HCV infected. Over the year, 4867 patients were screened in the primary health care center. Positive anti-HCV prevalence was 1.7% (2.1% in men and 1.3% in women). Blood transfusion and being a blood donor before 1993 also having tattoos were associated with higher odds of HCV infection. The study revealed the active participation of individuals in HCV screening.

## 1. Introduction

Hepatitis C virus (HCV) infection is a leading cause of chronic liver diseases such as liver cirrhosis and hepatocellular carcinoma [1]. According to the World Health Organization (WHO) estimates, globally viral hepatitis accounted for 1.34 million deaths in 2015 [2]. In the European Union/European Economic Area (EU/EEA), about 64,000 deaths were attributable to hepatitis B virus (HBV) and HCV in 2015, which included 55% of death from hepatocellular carcinoma and 45% from cirrhosis and other chronic liver diseases [3]. HCV infection is the predominant reason for liver transplantation [4].

At the beginning of 2020, the estimated global prevalence of viraemic HCV infection was 0.7%, corresponding to 56.8 million HCV-infected people [5]. In the European Union (EU), an estimated 3.24 million people lived with viraemic HCV infections in 2015 [6]. The prevalence of HCV infection varies between countries. Most Western and Central European countries have a relatively low prevalence of HCV infection; however, this infection is very prevalent in some neighboring countries such as Russia and Romania [6,7,8].

In Lithuania, representative population-based data on the prevalence of HCV infection are limited. A cross-sectional study of anonymous volunteers testing for HCV antibodies was conducted in five major cities of Lithuania in 2010 [9]. The seroprevalence of HCV found (2.78%) was probably too high because individuals with HCV risk factors might more actively participate in free anonymous testing for anti-HCV at the biggest supermarkets.

Data on diagnosed and newly diagnosed viraemic cases are also scarce in Lithuania. Individuals infected with HCV often do not feel any symptoms until a late stage of the disease, and up to half of them are unaware of their infection [10,11]. A study, which included 28 EU countries, estimated that only 36% of those with viraemic HCV infection were diagnosed [6].

In 2016, the 69th World Health Assembly approved the Global Health Sector Strategy to eliminate hepatitis infection by 2030, including a 90% reduction in new cases of chronic hepatitis C (CHC), a 65% reduction in mortality associated with HCV infection and diagnosis of 90%, as well as treatment of 80% of chronically infected patients, compared to the values reported in 2015 [12]. Achievement of the goals becomes a reality with direct-acting antiviral (DAA) therapies, which can cure the majority (more than 95%) of HCV-infected patients [13]. Early treatment would improve clinical outcomes and reduce viral transmission; however, a high proportion of HCV-infected people remain undiagnosed and are at risk of developing cirrhosis and hepatocellular carcinoma. National screening programs for HCV infection can identify asymptomatic infected individuals before they develop complications.

In 2022, the Lithuanian health authorities decided, as a first step of the screening program, to pay general practitioners (GPs) a special fee for a service of promoting and performing serological tests for HCV antibodies once per life in the population born from 1945 to 1994. Annual HCV testing by GPs is also planned for persons who inject drugs (PWID) or have human immunodeficiency virus (HIV). Such an initiative is one of the first in Central and Eastern Europe. The pilot study was carried out in the health care center of seaport Klaipeda, Lithuania.

For the development of national policies, reliable epidemiological data on the current situation are very important. This study aimed to assess trends in HCV-related mortality in the country and the prevalence of HCV infection among patients with liver diseases and evaluate possibilities of screening for HCV infection at a primary health care center.

## 2. Materials and Methods

### 2.1. Mortality Data

Data for the number of cause-specific deaths in the Lithuanian population from 2001 to 2020 were obtained from the Institute of Hygiene. The causes of death were coded according to the International Classification of Diseases 10th Revision (ICD-10). Starting in 2010, not only the underlying cause but also the contributory cause of death is coded. Three groups of death were analyzed: (1) deaths caused by CHC (B18.2 coded as an underlying cause), (2) deaths caused by other and unspecified liver cirrhosis coded as an underlying cause and CHC as a contributory cause (K74.6 with B18.2), and (3) deaths caused by hepatocellular carcinoma coded as an underlying cause and CHC as a contributory cause (C22.0 with B18.2). This grouping of death was chosen because CHC was most commonly associated with other and unspecified liver cirrhosis (K74.6) and hepatocellular carcinoma (C22.0) as a contributory cause of death.

### 2.2. Data on HCV Prevalence among Patients with Liver Diseases

The study was performed at the Hospital of Lithuanian University of Health Sciences Kauno Klinikos (hereinafter hospital Kauno Klinikos). This hospital is the largest multi-profile medical institution in Lithuania. Data were collected from the hospital electronic database on patients with liver cirrhosis, hepatocellular carcinoma and liver transplant who were consulted or treated in the hospital. If the patient visited the Kaunos Klinikos hospital multiple times, only data of the first visit were included. The study covered the years 2018–2021 for liver cirrhosis, 2018–2020 for hepatocellular carcinoma, and 2000–2021 for liver transplantation. Data on gender, birth date, visit date, and diagnosis of chronic HCV infection confirmed with a positive HCV-RNA test were collected.

### 2.3. Screening for Hepatitis C Virus Infection and Case-Control Study

Screening for HCV infection was carried out in the Jurininku Health Care Center located in seaport Klaipeda, Lithuania, from November 2020 to February 2022. The center provides primary health care services. In 2020, about 37,000 city inhabitants were registered with this health care center. Patients were invited to participate in the HCV screening by GPs during the visits. Additionally, regional media, social networking sites, and leaflets were used to inform about the possibility of being tested free of charge. Screening involved a blood test for the presence of antibodies in HCV (TOYO rapid test, Turklab Tibbi Malzemeler A.S., Turkey). Patients who tested positive were referred to a gastroenterologist or infectious disease doctor for further examination.

A case–control study was performed to identify risk factors for hepatitis C infection. All seropositive patients (case group) were invited to participate in a telephone interview, and 65 out of 81 (80.2%) responded. The cases were matched with controls by gender and age with a ratio of 1:2, selecting controls born in the same year as cases. The control group (n = 130) was randomly selected from seronegative patients and interviewed by telephone using the same questionnaire as the case group. The questionnaire included questions about the transfusion of blood products before 1993, being blood donors before 1993, injecting drug use, tattoos and being in prison for more than 3 months.

The study protocol was approved by Kaunas Regional Ethics Committee for Biomedical Research (protocol number BE-2-11). Written informed consent was signed by all participants.

### 2.4. Statistical Analysis

Yearly mortality rates per 100,000 people, age-standardized to the European standard population, were calculated [14].

The categorical variables were presented as proportions and compared using a χ^2^ test, Z-test with Bonferroni correction for multiple comparisons and Fisher’s exact test. Means were compared using Student’s *t*-test. The associations of possible risk factors with HCV infection were analyzed using univariable and multivariable logistic regression analysis.

Data analysis was performed using the statistical package IBM SPSS Statistics for Windows, Version 27.0 (IBM Corp.: Armonk, NY, USA, released 2020).

## 3. Results

Mortality from CHC as an underlying or contributory cause of death was lower among women than men (Figure 1, Appendix A). Including CHC as a contributory cause of death resulted in a nearly two-fold increase in mortality related to CHC. During the study period, no steady trend in mortality from all groups of death was observed. The years when mortality was highest or lowest varied between gender and cause of death. In 2020, mortality from CHC coded as an underlying cause and unspecified liver cirrhosis with CHC as a contributory cause decreased, while mortality from hepatocellular carcinoma with CHC as a contributory cause increased, especially in men, compared to previous years.

Analysis of medical documentation of patients visiting hospital Kauno Klinikos in 2018–2021 showed that 262 patients (146 men and 116 women) were diagnosed with liver cirrhosis. Among them, 106 (40.5%) patients were infected with HCV (Table 1). In men, the highest proportion (57.9%) of patients infected with HCV was found in the 50–59 years age group, while women showed increasing trends of the proportion from the youngest to the oldest age group.

Hepatocellular carcinoma was diagnosed in 197 patients (148 men and 49 women) visiting hospital Kauno Klinikos in 2018–2020 (Table 1). HCV infection was found in 43.2% of men and 69.4% of women with hepatocellular carcinoma. The highest proportion of HCV-infected men and women was in the age group 50–59 years, 76.2% and 76.9%, respectively. In older age groups of men with hepatocellular carcinoma, the prevalence of HCV infection was much lower (30.2% and 20.9%). No decrease in the prevalence of HCV infection among women was observed with age.

Between 2000 and 2021, 107 liver transplantations were performed in hospital Kauno Klinikos—48.4% out of 221 transplantations performed in Lithuania during the same period (Table 1). Liver transplantations were more common in men than women, 70 (65.4%) and 37 (34.6%), respectively. The proportion of HCV-infected patients was 41.4% in men and 27.3% in women. No statistically significant difference was found between age groups, possibly due to the small number of cases. Nevertheless, data showed that the prevalence of HCV infection was higher in younger age groups (up to 60 years) of men and had an increasing tendency with age in women.

From November 2020 to February 2022, 4867 patients (2363 men and 2504 women, 48.6% and 51.4%, respectively) were screened in the health care center in Klaipeda. Positive test results were found in 81 (1.7%) patients (Table 2). Of all screened patients, 4167 (85.6%) were born from 1945 to 1994, and 79 (1.9%) of them were seropositive. Seroprevalence of HCV antibodies was higher among men than women, 2.1% and 1.3%, respectively. In men, the highest HCV seroprevalence was found in the age groups 30–39 years (3.6%) and 40–49 years (3.5%). No statistically significant difference was observed between age groups in women.

In total, 195 individuals participated in the case–control study (53.8% men and 46.2% women). The mean age of participants in the case group was 52.9 (SD 9.3) years and mean of the control group was 53.2 (SD 8.8) years (*p* = 0.160). In the case group, possible risk factors for hepatitis C virus infection were more prevalent among men than women, except for blood transfusion before 1993 (Table 3). No differences were found in the control group. The prevalence of possible risk factors was higher among HCV seropositive than seronegative individuals. The proportion of seropositive participants who declared blood transfusion before 1993 was 5.3 times higher than seronegative. Among the cases, the prevalence of being blood donors before 1993 was 2.1 times higher and having tattoos 3.8 times higher compared to the control. Injection of illegal drugs was reported by 23.8% of HCV-positive patients; 27.0% of seropositive patients were in prison for more than three months. The latter risk factors were not reported by the participants of the control group. The differences in the prevalence of risk factors between seropositive and seronegative individuals were higher in men than women.

The univariable logistic regression analysis showed that blood transfusion before 1993, being blood donors before 1993, and having tattoos increased the odds of seropositivity of HCV statistically significantly (Table 4). The multivariable logistic regression analysis, which included all five variables, confirmed that the odds of HCV antibodies in patients who reported blood transfusion before 1993 were 6.81 times higher compared to those who did not have a blood transfusion or had it later. HCV antibodies were found more often among patients who were blood donors before 1993 compared to those who were not donors or were donors later (OR 4.59; 95% CI 2.08–10.15). Having a tattoo increased the odds of HCV seropositivity by 6.46 times.

## 4. Discussion

Evaluation of HCV-related disease burden is important to support the elaboration of the national strategy to eliminate HCV infection. Our study analyzed trends in HCV-related mortality and assessed the HCV prevalence among patients with liver cirrhosis, cancer and liver transplant patients. The seroprevalence of HCV antibodies was established among patients registered at the primary health care center, and an association between HCV infection and risk factors was evaluated.

HCV-related mortality has not changed significantly in Lithuania over the last ten years. In 2020, a lower mortality rate from CHC as an underlying cause was observed. However, it is difficult to interpret these changes due to redistributions of death causes and coding problems due to the COVID-19 pandemic. Some upward trends in mortality from hepatocellular carcinoma with CHC as a contributory cause may be related to a better diagnosis of HCV infection in recent years. The studies, carried out in the United Kingdom and Australia, showed a reduction in the number of deaths from HCV-related liver diseases after 2014, when new treatments with DAAs were introduced [15,16,17]. DAA therapy is applied in Lithuania. All HCV-infected patients are eligible for reimbursement of treatment [18]. Unfortunately, most of them are still undiagnosed and untreated. In our study, HCV-related mortality was higher in men than women. Similarly, analysis of death data in EU/EEA countries showed a more than three-fold higher mortality rate from hepatocellular carcinoma caused by CHC among men than women [3]. The studies demonstrated the high impact of HCV infection on all-cause mortality and, in particular, liver-related mortality [3,19,20].

Estimation of the mortality attributed to HBV and HCV infections is challenging because these infections are often underreported in mortality records [15,21,22]. In the cohort study carried out in England, HCV was not mentioned on the death certificate of 45% of HCV-infected patients [15]. Our and other studies showed that adding CHC as a contributory cause to liver cirrhosis and cancer as an underlying cause increases the mortality rates from HCV-related diseases [15,20]. Simmons et al. found that HCV was recorded as a contributory cause of death for 28.4% of underlying causes [20].

There is a lot of evidence that HCV infection is associated with chronic liver diseases [1,23,24,25,26]. HCV induces liver fibrosis and cirrhosis. The incidence of cirrhosis ranges from 15% to 35% during the 25–30 year period after HCV infection. The risk of hepatocellular carcinoma increases with the fibrosis stage. In patients with cirrhosis, hepatocellular carcinoma develops at an annual rate of 1%–4% [1,24]. Analysis of medical documentation of patients in Kaunas Klinikos hospital revealed that 40.5% of patients with liver cirrhosis, 49.7% with cancers and 36.9% of transplant patients were HCV infected. The highest proportion of HCV-infected patients with liver diseases was in the age group 50–59 years. Other authors found that since the introduction of DAA, HCV-related cirrhosis significantly decreased as an indication of liver transplantation; however, the short time from HCV diagnosis to liver transplantation suggests that late diagnosis of HCV infection remains a serious problem [27,28].

Screening strategies vary between different countries. In some countries, screenings enrolled specific populations, such as blood donors, persons with other risk factors or clinical patients, who are not representative of the general population [29]. Recent studies assessed that screening of the general population is cost-effective compared to the screening of high-risk individuals or birth cohort-based screening alone [30,31]. The Lithuanian health authorities decided to start with birth cohort-based screening for inhabitants born from 1945 to 1994 (testing once per life) and annual HCV testing of PWID and patients with HIV. The pilot study in Klaipeda confirmed that most (97.5%) anti-HCV-positive cases identified were in a selected birth cohort. For one year, almost 5000 patients were screened in the primary health care center. Whereas screening was performed during the COVID-19 pandemic, when patient visits to the health care center were restricted, and online doctor consultations were offered, patient participation in screening was good enough.

In our study, the seroprevalence of HCV antibodies (1.7%) was lower than in the study carried out in Lithuania in 2010 (2.78%) [9]. Representativeness of the latter study was lower because it involved volunteers, and testing for anti-HCV was performed at the biggest supermarkets free of charge. Higher participation of people at HCV risk and overestimation of the anti-HCV prevalence was very likely. The seroprevalence of HCV antibodies identified in our study was similar to that in neighboring Baltic countries: in Estonia, 1.5–2.0% [32] and Latvia, 2.4% [33]. A similar proportion of seropositive people were found in Central European countries [8,18]. Analysis of the national surveillance system data indicated that the number of acute HCV infection cases diagnosed and registered in Lithuania from 2005 to 2018 showed a decreasing trend (from 2.0 cases per 100,000 in 2005 to 0.9 cases per 100,000 in 2018) [34]. However, it must be emphasized that a large number of such cases were not captured by the current national surveillance system.

Data of our case–control study demonstrated that blood transfusion and being a blood donor before 1993, as well as having tattoos, were associated with higher odds of HCV infection. These risk factors were most prevalent among HCV-infected individuals, especially men. We were not able to calculate the odds ratios of HCV infection for illicit drug users and participants who were in prison for more than three months because such risk factors were not found in the control group. However, the high proportion of anti-HCV-positive individuals who reported injection drug use suggests that this risk factor is of great concern, especially among young people. The previous study of CHC patients carried out in Lithuania found that the main routes of HCV transmission were intravenous drug use and tattoos [35]. Analysis of the national surveillance system data also confirmed that injection drug use was the most commonly recorded transmission route for acute HCV cases in Lithuania; however, information about the possible transmission routes was not provided for 56.2% of all the analyzed cases [34]. Evidence suggests that the relative impact of the different routes of HCV transmission has changed over the last decades [36,37,38]. Before screening assays became available, most HCV infections were iatrogenic caused by transfusions with infected blood or unsafe invasive medical and surgical procedures. So far, the majority of diagnosed HCV patients in Lithuania have been infected through blood transfusions and blood donation before 1993. Meanwhile, new HCV infections are related to illicit drug use. Estimated anti-HCV prevalence among injection drug users was 60% or higher in half of the European countries [36]. Thus, including the annual testing of injection drug users in the Lithuanian screening program was the right decision.

Several limitations of the study should be mentioned. It is likely that the analysis of mortality did not capture all death caused by HCV because of underreporting of this infection in mortality records. We expect that adding CHC as a contributory cause to liver cirrhosis and cancer as an underlying cause helped us to estimate the mortality rate from HCV-related diseases more accurately. Moreover, there is no reason to believe that underreporting has changed over time and might have an impact on mortality trends. Analysis of medical documentation on the proportion of liver diseases caused by HCV was performed only in one hospital. However, Kaunas Klinikos hospital is one of the largest in Lithuania, visited by a high proportion of patients with liver disease. Furthermore, the pilot study of screening was organized only in one primary health care center. Although enough people have been screened, the seroprevalence of HCV infection can be not representative of the whole Lithuanian population. Information provided by the participants during a telephone interview on potential risk factors for HCV infection may be biased because some of them may be reluctant to tell the truth about sensitive personal matters. We did not possess data regarding what proportion of anti-HCV positive screened subjects was HCV-RNA positive. The participants with positive anti-HCV serologic tests were referred to a gastroenterologist or infectious disease doctor for HCV-RNA test and treatment if required. The specialists do not directly inform the GPs about the test results. This information is provided in an electronic health system, but its availability at the individual level is limited due to data protection. In Lithuania, extrapolated viraemic rate of 66% from the anti-HCV positive cases is used [7]. According to this rate, 53 out of the 81 anti-HCV positive subjects identified in our study would be HCV-RNA positive, and the prevalence of HCV-RNA positivity would be 1.1%, the same as reported by Polaris Observatory for the Lithuanian population [7]. More reliable data will be available when HCV testing is started at the national level in Lithuania in May 2022.

In conclusion, our study provided new data on the HCV epidemiological situation in Lithuania. They revealed the active participation of individuals registered with primary health care centers in HCV screening performed by GPs. Population-based screening may be an important tool for assessing the number of infected with HCV people and enabling estimation of the future disease burden. The screening strategy chosen by Lithuanian health authorities will allow the identification of the majority of HCV-infected people. Updated epidemiologic data are necessary to support the elaboration of the national strategy for elimination of HCV infection and achievement of the WHO targets by 2030, including national population-based screening programs, appropriate preventive measures to reduce the incidence of HCV infection and effective treatment of HCV infected people in the early stage of the disease.

## Figures and Tables

**Figure 1 viruses-14-01192-f001:**
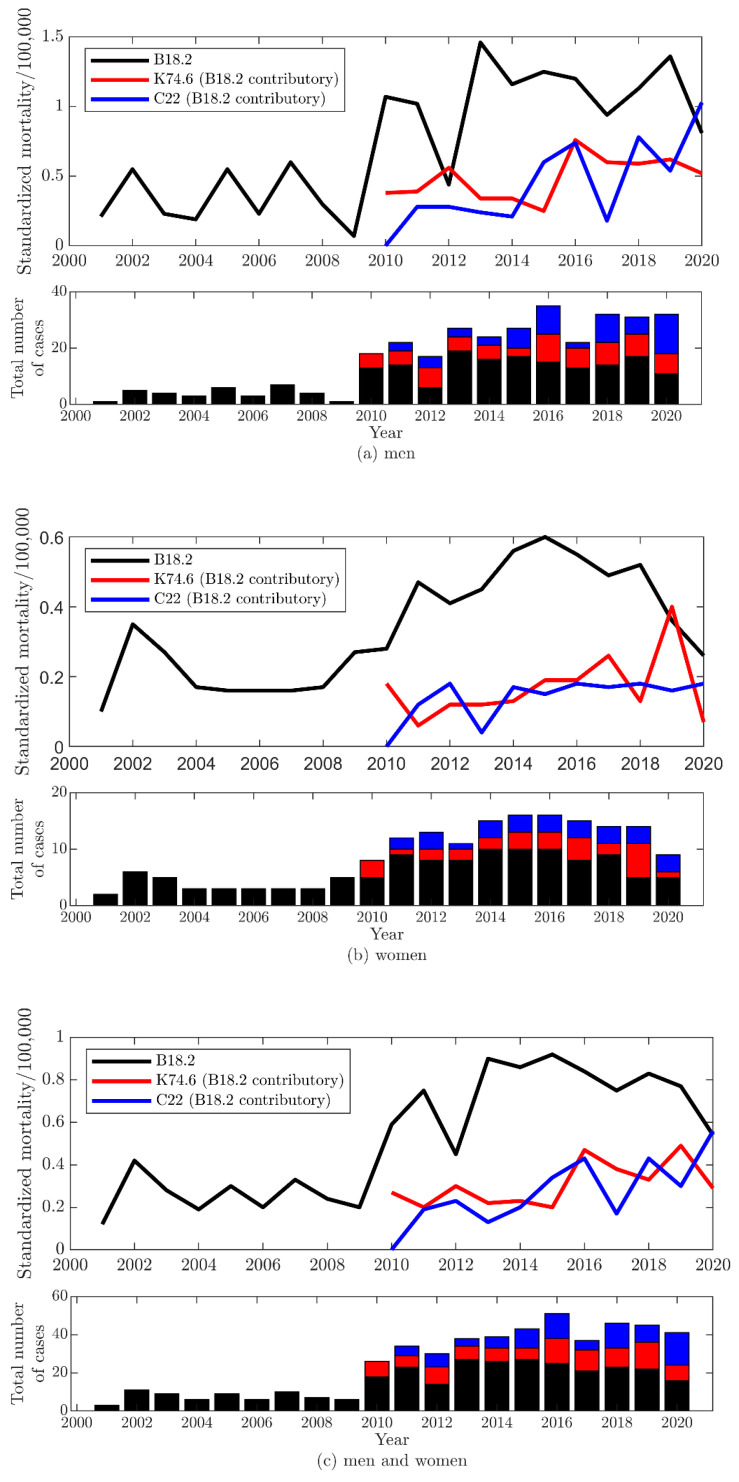
Sex-stratified and total annual age-standardized mortality rates per 100,000 people for chronic viral hepatitis C (CHC) as an underlying cause, unspecified liver cirrhosis and hepatocellular carcinoma with CHC as a contributory cause: (**a**) men; (**b**) women; (**c**) men and women.

**Table 1 viruses-14-01192-t001:** Prevalence of chronic hepatitis C virus infection among patients with liver cirrhosis, hepatocellular carcinoma and liver transplant patients, according to gender and age (data of hospital Kauno Klinikos).

Gender	Age Groups (Years)					*p*-Value between Age Groups
	Under 50	50–59	60–69	70 and More	Total	
Liver cirrhosis (n = 262)						
Men n (%)	17 (37.0)	33 (57.9) **	7 (23.3)	2 (15.4)	59 (40.4)	0.002
Women n (%)	5 (21.7) *	17 (37.0)	12 (46.2)	13 (61.9) *	47 (40.5)	0.047
Total n (%)	22 (31.9) *	50 (48.5) *	19 (33.9)	15 (44.1)	106 (40.5)	0.108
Hepatocellular carcinoma (n = 197)						
Men n (%)	7 (70.0)	32 (76.2) *	16 (30.2)	9 (20.9) *	64 (43.2)	<0.001
Women n (%)	2 (66.7)	10 (76.9)	10 (66.7)	12 (66.7)	34 (69.4)	0.925
Total n (%)	9 (69.2)	42 (76.4) *	26 (38.2) *	21 (34.4)	98 (49.7)	<0.001
Liver transplantation (n = 107)						
Men n (%)	12 (40)	16 (43.2)	1 (33.3)	0	29 (41.4)	0.925
Women n (%)	2 (15.4)	4 (33.3)	3 (42.9)	0	9 (27.3)	0.701
Total n (%)	14 (32.6)	20 (40.8)	4 (40.0)	0	38 (36.9)	0.790

* *p* < 0.05 between marked groups (*z* test with Bonferroni correction), ** *p* < 0.05 compared to other age groups (*z* test with Bonferroni correction).

**Table 2 viruses-14-01192-t002:** Seroprevalence of hepatitis C virus infection among the people screened at the primary health care center in Klaipeda.

Age Groups (Years)	Men		Women		Total		*p*-Value between Men and Women
	Total Screened	Seropositive	Total Screened	Seropositive	Total Screened	Seropositive	
	n	n (%)	n	n (%)	n	n (%)	
<30	229	1 (0.4)	183	1 (0.5)	412	2 (0.5)	1.000
30–39	251	9 (3.6)	262	1 (0.4)	513	10 (1.9)	0.010
40–49	520	18 (3.5)	333	5 (1.5)	853	23 (2.7)	0.127
50–59	562	15 (2.7)	668	13 (1.9)	1230	28 (2.3)	0.446
60–69	487	6 (1.2) *	550	9 (1.6)	1037	15 (1.4)	0.615
≥70	314	0	508	3 (0.6)	822	3 (0.4) *	0.291
*p*-Value	0.001		0.192		0.001		-
Total	2363	49 (2.1)	2504	32 (1.3)	4867	81 (1.7)	0.033

* *p* < 0.05 compared to age groups 40–49 and 50–59 (z test with Bonferroni correction).

**Table 3 viruses-14-01192-t003:** Prevalence (%) of possible risk factors for hepatitis C virus infection in seropositive and seronegative individuals.

Risk Factor	Men (n = 105)			Women (n = 90)			Total (n = 195)		
	Anti HCV+ (n = 35)	Anti HCV− (n = 70)	*p*-Value	Anti HCV+ (n = 30)	Anti HCV− (n = 60)	*p*-Value	Anti HCV+ (n = 65)	Anti HCV− (n = 130)	*p*-Value
Blood transfusion before 1993	17.1	1.4	0.006	23.3	6.6	0.036	20.0	3.8	<0.001
Blood donors before 1993	60.0	29.0	0.002	33.3 *	14.8	0.04	47.7	22.3	<0.001
Tattoo	60.0	10.1	<0.001	17.2 *	11.5	0.452	40.6	10.8	<0.001
Injection of illegal drugs	40.0	-		3.6 *	-		23.8	-	
Being in prison for more than 3 months	45.7	-		3.6 *			27.0		

* *p* < 0.05, compared with men in anti HCV+ group; anti HCV+ are seropositive; anti HCV− are seronegative

**Table 4 viruses-14-01192-t004:** Odds ratios (95% CI) of seropositivity of hepatitis C virus infection by possible risk factors (logistic regression analysis).

Risk Factor	Univariable Analysis			Multivariable Analysis *		
	OR	95% CI	*p*-Value	OR	95% CI	*p*-Value
Blood transfusion before 1993	6.25	2.12–18.42	0.001	6.81	2.00–23.23	0.002
Blood donors before 1993	3.18	1.68–6.01	<0.001	4.59	2.08–10.15	<0.001
Tattoo	5.67	2.67–11.95	<0.001	6.46	2.81–14.82	<0.001
Women vs. men	0.97	0.53–1.76	0.930	0.75	0.36–1.58	0.454
Age	0.98	0.95–1.01	0.160	0.96	0.92–0.99	0.047

* All five variables are included; OR-odds ratio; CI-confidence intervals.

## Data Availability

The data presented in this study are available on request from the corresponding author. The data are not publicly available due to ethical issues.

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
