# Peer review of "Hepatitis C Virus Epidemiology in Lithuania: Situation before Introduction of the National Screening Programme"

_viruses, 2022, doi:10.3390/v14061192_

Round 1

Reviewer 1 Report

introduction:  (39) liver transplantation data are from 2008-2017 and it is not correct to use term "currently".

results: (158) "liver cancer" , table 2 "liver cell carcinoma", table 1 "hepatocellular carcinoma" - please use  the same term in the text

results: table 2 please add %

results: please HCV RNA prevalence in 81 antiHCV positive patients

Reviewer 2 Report

In their work Ciupkeviciene and colleagues presented data on Hepatitis C virus epidemiology in Lithuania, presenting data on mortality due to chronic hepatitis C (CHC) with or without hepatocellular carcinoma (HCC) over time, as well as data on the prevalence of CHC or HCV-Ab seropositivity in a group of patients (with or without CHC) attending a specific hospital in the country.

The paper could fill a gap of reliable information on the burden of HCV infection in Lithuania and provide important information to benchmark the effect of a promoted HCV screening program in the general population.

The paper reports three aspects of HCV epidemiology, not completely related to each other. There are some aspects of concern (mainly methodological) that could be better explained to have a clear judgment of the conclusion made by the authors.

Major points of concern:

It is hard to follow the considerations derived from data shown in Table 1. I would advise authors to show the same data in a three panel figures were trend of age-standardized rates for men, women and the total population are reported as well as counts of cases below the single year shown in the figure; it would be easier for the reader to judge an overall stability of the mortality rates.

While it seems for CHC underlying cause that no trend has been observed over time, it seems however that for unspecified cirrhosis with CHC as contributory cause as well as for HCC mortality with CHC as contributory cause, a doubled rate has been observed in the last 5-years (in men and overall). This augmented (even if arising from a small number of cases) was not highlighted neither discussed. A possible explanation of this difference could be explained more likely by the not negligible number of underdiagnosed HCV infections that has probably decreased over time, and not because an increasing mortality rate over time. It would be important for example to have also data on the trend of mortality due to chronic unspecified viral hepatitis (B18.9) that presumably decreased over time for the abovementioned improvement of HCV diagnostic capabilities, especially since the introduction of DAA from 2015.

In table 1, data on age-standardized rates should be shown as Rates not Ratio. Moreover, the three columns showing data of the total of men+women for the three causes of death considered are likely referred to rates not absolute numbers (n).

It is not clear if the patients enrolled in the prevalence study found to be seropositive at HCV rapid test, have been later confirmed by standard laboratory test once linked to care to a gastroenterologist. It was not stated also if all 81 cases were chronically infected or only (as it seems) seropositive for HCV. This last aspect will need to be better clarified since the previous analysis shown all referred to chronic HCV infection and its sequelae. We can consider the chronic HCV infection would be identified in 3/4 of them, that probably should be the cases considered in the case-control study or at least in a subgroup of anlaysis.

Authors stated that in the following case-control study, all cases were matched by gender and age (exact age or for example +/- 1 or 2 years?, please specify) with a case:controls ratio of 1:2.

Overall 65 (out of 81) cases were interviewed matched with twice the number of controls (130). However, in Table 4 there is an odd number of cases and matched controls in men (35:69) and in women (30:61) with one more female control than expected. This error of matching probably was evident only after collecting data and could the otherwise unconceivable difference in proportion of males and female in cases and controls, that according to the study design have to be indentical (the same also for age differences).

In the analysis on logistic regression it is not clear if the model included five variables at the same time considering the three important (already known) factors identified to be associated with HCV-seropositivity in the seroprevalence study (namely: blood transfusion of donation before 1993 and tattoos), with age and gender.

Finally, authors cited some papers showing that after the introduction of DAAs, the proportion of HCV-related liver transplants decreased (refs. 27-28, page 7 lines 251-253). The results reported in Table 2 are an overall prevalence of all liver transplant performed in more than twenty years (less than 5 a year) that does not allow to make any consideration of any trend over time. For this reason the phrase “However, the short time from HCV diagnosis to liver transplant suggests that late diagnosis of HCV infection is an important problem”, it is not only unclear but neither based on the results reported.

Minor points:

On page 2, line 63, probably there is a typo on the acronym of Direct-acting antiviral (DDA instead of DAA).

In table 2, I would advise authors to better identify younger patients reporting the label “lower than 50” or “<50” or “up to 49”, in order to avoid overlapping with the next class (50-59). Moreover, I would advise to be consistent with Table 1 referring to hepatocellular cancer (C22.0) instead of liver cell cancer, unless a different definition of the disease has been used.

The paper needs a revision of the English form.

For the above-mentioned points, I would suggest this paper not be considered in the present form acceptable for publication, considering that a major (mainly on methodology, conclusions and sub analysis to be performed) revision could heavily improve the quality of the paper.

Round 2

Reviewer 2 Report

The new version of the paper addressed all highlighted points needing more attention.

In table 2 please do not add the symbol percentage after the single proportion indicated since it is stated that values are reported as counts and percentage [n (%)]. This will make the table consistent to Table 3.

Author Response

Dear reviewer,

Thank you for your comment: ‘In table 2 please do not add the symbol percentage after the single proportion indicated since it is stated that values are reported as counts and percentage [n (%)]. This will make the table consistent to Table 3.’

Following your advice, we removed the percentages in Table 2.

Thank you once again for reviewing our manuscript.

Egle Ciupkeviciene